# Managing Local Health System Interdependencies: Referral and Outreach Systems for Maternal and Newborn Health in Three South African Districts

Helen Schneider [1,*], Solange Mianda [1], Willem Odendaal [2,3] and Terusha Chetty [2,4]

1   SAMRC Health Services to Systems Unit, School of Public Health, University of the Western Cape, Robert Sobukwe Road, Bellville, Cape Town 7535, South Africa; smianda@uwc.ac.za
2   HIV and Other Infectious Diseases Research Unit, South African Medical Research Council, Francie van Zijl Drive, Parow Valley 7505, South Africa; willem.odendaal@mrc.ac.za (W.O.); terusha.chetty@mrc.ac.za (T.C.)
3   Department of Psychiatry, Stellenbosch University, Francie van Zijl Drive, Tygerberg, Western Cape, Cape Town 7505, South Africa
4   Discipline of Public Health Medicine, School of Nursing and Public Health, University of KwaZulu-Natal, Durban 4041, South Africa
*   Correspondence: hschneider@uwc.ac.za

**Abstract:** In complex health systems, referral and outreach systems (ROS) are formal strategies for connecting and managing interdependencies between facilities in service delivery pathways. Well-functioning maternal and newborn ROS are critical to successful outcomes, and therefore, a good lens through which to examine the management of local interdependencies. We conducted a qualitative study of maternal–newborn ROS, involving interviews with 52 senior, middle, and frontline managers, in three health districts of three different provinces in South Africa. We analyse the differences in functioning of ROS as an interplay of setting (urban, rural), individual facility strengths and weaknesses, the quality of emergency medical services (EMS), and the wider provincial strategic and organisational context. ROS are strengthened by sub-district governance arrangements that recognise and enable connectedness—in particular, between primary health care and district hospital services; by informal, day-to-day communication and collaboration across levels and professions; and by hybrid clinical–managerial players as system brokers and systems thinkers. We also identify leverage points, places where small shifts could have wider system effects, most notably in the design and functioning of EMS, and in addressing small, but significant bottlenecks in supply chains in lower level facilities that negatively impact the system as a whole.

**Keywords:** interdependence; interdependencies; leverage points; referral; outreach; system; district health systems; sub-district health systems



## 1. Introduction

Health systems are complex organisations that require coordinated action between diverse players, levels, and processes to achieve health outcomes [1]. The performance of health systems rests not only on the functioning of individual system elements, but also on "the relationships, connections and interactions among parts of a complex system" [2] referred to as "interdependencies" [2,3]. Interdependence is also framed as the "problem of many hands", denoting the multiple actors, individual and collective, in health systems that collectively produce outcomes (positive or negative), and for which no single actor can be held responsible [4].

Maternal and newborn health (MNH) care is a classic example of health system interdependence, requiring preventive action in primary health care (PHC), early detection of obstetric complications, initiation of referral procedures, functioning emergency medical services (EMS), and appropriately resourced higher level facilities able to provide

emergency care [5]. If any level is dysfunctional, the impact is felt throughout the system, resulting in delays in timely and appropriate patient management and avoidable deaths. The three delays model, an explanatory framework for maternal mortality, outlines the three interfaces and phases of delay that affect reaching and receiving adequate maternal care at different levels of the health system [6,7]. These delays include seeking help for an obstetric emergency, reaching an appropriate facility, and receiving adequate quality of care. Poorly functioning referral systems are frequently an attributable factor in maternal and newborn mortality [8–13].

Referral and outreach systems are mechanisms that connect different elements of the health system, typically across organisational boundaries. These systems facilitate integration and coordination between levels of care, across specialties, and within teams, seeking to ensure optimal access and quality within heterogenous care systems. Referral concerns the movement of patients upwards, downwards, or laterally through care pathways [14], and outreach, which is the flow of knowledge and support from (central) specialist to (more peripheral) generalist cadres and levels [15]. They involve largely the same health system actors, are enabled by the same underlying governance and system capacities, and are mutually reinforcing.

Murray and Pearson [5] proposed the following as dimensions of a well-functioning maternal health referral system: referral policy/strategy to guide the functions and links between levels; adequately resourced facilities along the referral pathway; active collaboration and coordination between and across referral levels; formal and informal systems of communication; planned and emergency transport arrangements; and mechanisms of accountability, supervision, monitoring, and evaluation. Specialist or clinical outreach (as distinct from community outreach) includes a variety of activities that combine aspects of clinical care, continuing professional education, and clinical governance [15]. Specialist outreach can also support system strengthening by advocating for the needs of health care providers to higher levels, and participating in decision-making in the procurement of supplies and equipment, or staff appointments.

To function effectively, referral and outreach systems need to be championed by district managers and embedded in local relational contexts and management systems such as skills development plans, clinical governance, and wider accountability mechanisms [16].

While recognised as important, there is little systematic understanding of the design and functioning of referral and outreach in district health systems, especially when considered from a systems lens. This paper reports on a qualitative study of referral and outreach systems for MNH care in three districts of three South African provinces and what these systems reveal about the local management of health system interdependencies.

*Study Setting*

Health care in South Africa is provided in a plural health system of private and a public sectors, with the latter providing care to the majority (+80%) of the population [17]. The health system consists of a national and nine provincial departments of health, further divided in 52 districts, conterminous with the boundaries of district municipalities. The public health sector provides free comprehensive and accessible PHC within the district health system (DHS) in a continuum of care from community-based to PHC to district and regional hospitals following a hierarchical referral system (Figure 1). MNH care is exempt from user fees at all levels of the health system, and South Africa has relatively low levels of out-of-pocket expenditure for health care [18].

In 2012, as part of a wider set of strategies to strengthen PHC and accelerate achievement of the Millennium Development Goals, the national Department of Health introduced district-based clinical specialist teams (DCSTs) to support maternal, newborn, and child outreach services [19]. The teams were expected to have seven members, including a nurse–doctor dyad representing three key disciplines, namely family medicine, obstetrics, and gynaecology/midwifery and paediatrics, and an anaesthetist [19]. Their primary role was to oversee the quality of service delivery and ensure effective resource management

through clinical governance at the district level [20]. However, DCSTs were considered a costly and somewhat vertical programme [21], and are being phased out in parts of the country in favour of alternative models, including in one of the study provinces.

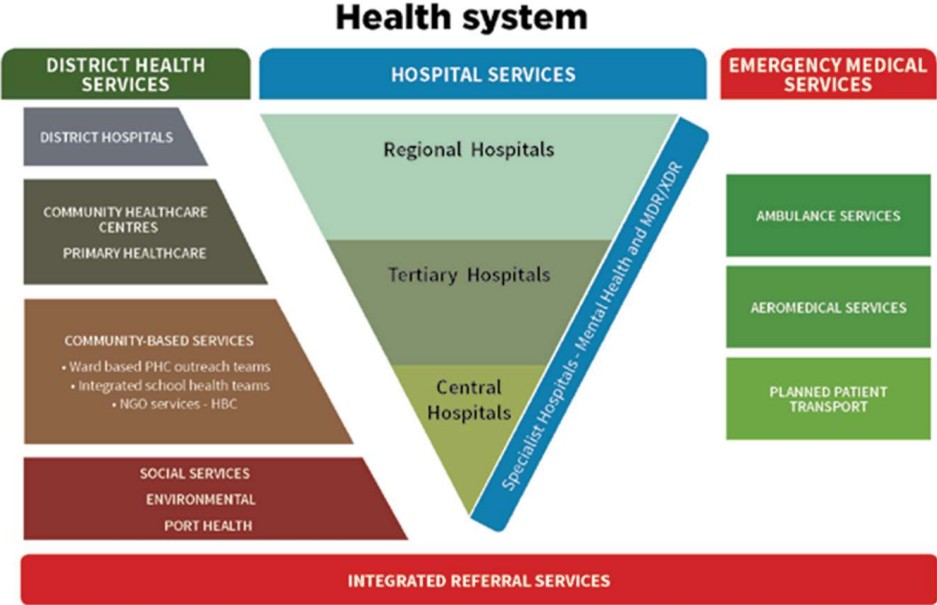

**Figure 1.** South Africa's public health sector service delivery system (source: [14]).

In 2020, the National Department of Health published a National Referral Policy [14], outlining the roles of different authorities and facilities (Figure 1) in the referral pathway. A 2021 Maternal, Perinatal and Neonatal Health Policy recommended the establishment of "functional and effective referral pathways" at district level "tailored according to the catchment area considering the availability of the next level of expertise and irrespective of health district demarcations" [22].

South Africa's EMS system plays a crucial role in supporting the service delivery referral platform by facilitating transport to the most suitable level of care. EMS consists of ambulance services, aeromedical services, and planned patient transport/transfer between levels of care [14]. The EMS system is regulated by the National Health Act (2003) and divided into three levels of response: basic life support, intermediate life support, and advanced life support, which correspond with levels of EMS personnel training [23]. The EMS system is managed provincially through central EMS call centres, which receive and triage requests and dispatch ambulances allocated to district and sub-district stations.

This study forms part of a wider project, referred to as "Mphatlalatsane", implemented to improve maternal and neonatal health (MNH) outcomes through quality improvement approaches at the facility and district level in three South African provinces [24]. District 1 is a metropolitan district; Districts 2 and 3 are mostly rural districts, consisting of a mix of small and medium-sized towns, farms, and mining areas. The public sector is the main provider of health care in these districts through a network of district-managed services and facilities (Table 1). The district authority is responsible for all services in District 2, including the regional hospital, whereas in Districts 1 and 3, regional and tertiary hospitals report through a separate line to provincial authorities. District 2 is the most rural of the districts, with the tertiary referral hospital some 100 kms away from the regional hospital. In 2020, all three districts scored below the national average (58.3/100) of a 'Universal Health Coverage (UHC) service index', a composite of 15 programmatic, socio-economic, and health system capacity indicators developed in South Africa to compare district performance [25].

**Table 1.** Profile of study districts (source: [25]).

|  | District 1 (D1) | District 2 (D2) | District 3 (D3) |
| --- | --- | --- | --- |
| Population | 1,320,576 | 1,238,398 | 1,743,182 |
| Geography | Urban Metro | Rural | Rural |
| Sub-districts | 3 | 5 | 4 |
| Health facilities |  |  |  |
| Tertiary hospital (TH) | 2 | 0 | 1 |
| Regional hospital (RH) | 1 | 1 | 2 |
| District hospital (DH) | 1 | 6 | 8 |
| Community health centres (CHC) | 9 | 8 | 15 |
| PHC clinics | 39 | 97 | 110 |
| Other hospitals * | 4 | 3 | 6 |
| UHC coverage index (out of 100) | 53.3 | 56.8 | 49.8 |

* TB, mental health.

## 2. Methods

### 2.1. Study Design

This qualitative study is one component of the Mphatlalatsane Project evaluation [26], which aims to document the meso (sub-district and district) and macro-level (provincial and national) contextual factors that influenced the implementation of the interventions. As part of this, we explored the models and functioning of referral and specialist outreach systems with health system managers and partners in the three provinces.

### 2.2. Study Population and Sample

We purposefully selected key informants from amongst provincial, district, sub-district and hospital managers, programme coordinators, and Mphatlalatsane implementing partners. These included provincial managers responsible for overall strategic direction (top managers), programme managers steering MNH across all levels, district executive managers (DEMs) and their teams, senior hospital executives in regional hospitals (middle managers), and sub-district PHC managers and district hospital managers from one sub-district per district (frontline managers).

Between September 2021 and February 2022, 52 managers and partners were interviewed in the three districts, as shown in Table 2.

**Table 2.** Profile of managers/partners interviewed.

| Level | District 1 | District 2 | District 3 | Total |
| --- | --- | --- | --- | --- |
| Provincial (Pr) | 1 | 4 | 1 | 6 |
| District (D) | 4 | 5 | 10 | 19 |
| Sub-district/facility (S) | 4 | 8 | 9 | 21 |
| Partners (Pa) | 1 | 2 | 3 | 6 |
| Total | 10 | 19 | 23 | 52 |

### 2.3. Data Collection and Analysis

We conducted mostly individual and some group semi-structured interviews tailored to particular roles of interviewees. Questions covered an array of themes, including: views on referral policy, functionality of referral systems, relationships between facilities (PHC, and district and regional hospitals), the role of EMS in referral, and specialist outreach and support systems. Interviews were mostly face-to-face, conducted in field visits to the three provinces, with some follow-up virtual interviews. The duration of interviews ranged between 45 and 60 min. Participation in the study was voluntary and all interviews were conducted after obtaining signed consent from participants. Interviews were recorded and

transcribed verbatim, ensuring respondent anonymity. Preliminary study findings were presented to the different districts for validation in four report back workshops.

Data analysis involved immersion and familiarisation by two of the authors (SM and HS), including re-reading transcripts and listening to the audio files, then extraction and coding of data on referral and outreach systems. We adapted Murray and Pearson's criteria [5] for analysing maternal referral systems into the following broad domains for coding: presence of a referral policy/strategy; adequately resourced sending and receiving facilities; functioning inter-facility transfers by EMS; and communication and collaboration between levels of care. For outreach, we documented the overall model in the three provinces and three roles: clinical care, training and mentoring, and audit/monitoring and evaluation. We considered the influence of governance and leadership on both referral and outreach. The raw data were categorised into the main constructs and assigned codes, then grouped into themes.

Ethical clearance to conduct the study was provided by the University of the Western Cape's Biomedical Research Ethics Committee (BM19/10/16), and the respective Provincial Research Committees. As provincial and district contexts are dynamic and shifting, and some of our findings may no longer pertain, we elected to anonymise both provinces (P1–3) and districts (D1–3) in reporting findings. Further, to maximise individual confidentiality, we assigned generic attributions to quotes indicating the district (D1–3) and the general category of respondent (as outlined in Table 2).

## 3. Results

We begin by reporting on the functioning of MNH referral and specialist outreach support systems in the three districts through the eyes of local actors. We then consider key governance factors enabling and constraining these local systems and draw out lessons for the management of interdependencies in local health systems.

### 3.1. Referral Systems

The perceived functioning of referral systems in the three districts lay on a continuum from mostly functional in D1, to improving *'but not 100% smooth'* in D2, to *'a massive challenge'* in D3. These differences arose from a complex interplay of the settings (urban, rural), strengths and weaknesses of individual facilities in the referral pathway (including facilities beyond the district), EMS functioning, the nature of relationships between actors in the pathway, local agreements, and the wider organisation of services and governance processes. This complexity is illustrated in the description of events leading to a maternal death in Box 1.

**Box 1.** A maternal death.

> *A senior provincial clinician recounted how a junior doctor in a district hospital. "... had a patient who was bleeding in theatre, he did everything, but the patient continued to bleed. We said, okay, ... tie the uterus and then call the ambulance ... He tried to call these people who were nearby who were senior, he couldn't get hold of them, but for him to sit and see ... the patient ... move from pink to pale to death ... He's supposed to have a senior person when he's in trouble, but the only senior person he can get is in [the capital city] on the phone. The second thing is the ambulance must quickly be able to come. When the ambulance comes, you find that they didn't bring the correct experienced person... Now they have to ... wait for the advanced person, the patient is still continuing to bleed.... He is waiting for the blood, in the fridges there is only two bloods. The other blood he can only get is in the blood bank two hours away. When he called ... they said the driver is going and an hour later you call the driver, and he says, no, no, I didn't get the message, was I supposed to go there?"* (Pr, D2)

Supplementary Table S1 provides a detailed inventory of referral themes along the dimensions outlined in the methods for each district, and explored further in the narrative below.

### 3.1.1. Referral Policy/Strategy

All three districts reported provincial referral policies, but these policies remained *'desk top'* (S,D3) unless they were (re)-negotiated, adapted, and elaborated into standard operating procedures for specific district and sub-district contexts. Local agreements were required, firstly, to manage cross-border flows of patients between sub-districts, districts, provinces, and even countries, as natural catchment areas did not always correspond with sub-district and district demarcations; secondly, on the allocation of roles, such as where uncomplicated maternal deliveries would take place and whether regional hospitals could provide district hospital services to surrounding communities; and thirdly, on bypassing when the requisite expertise was not available at the next level of the pathway. These were raised as issues of concern particularly in D2 and D3. Provincial authorities introduced mechanisms (including service level agreements and regular forums) to address these challenges in D2, while D3 relied mostly on informal arrangements or ad hoc facilitation by partners to resolve problems. As recounted by a clinical manager in D3: *"We were struggling when we called X [regional hospital name], X would say no, no you refer to Y [regional hospital name] and you call Y and they would say no, no refer to X... Then that is when Mphatlalatsane came in, lucky enough, and the district also came in... then it was sorted out"* (S,D3).

### 3.1.2. Adequately Resourced Facilities

Referral systems depended not only on agreed roles in the referral pathway, but also on the capacity to fulfil these roles. In all three districts, interviewees described imbalances in the distribution of resources and capabilities. This took different forms. One sub-district of D1 had a *"distorted service delivery platform"* (D, D1) of regional and tertiary facilities, but no district hospital beds, and a legacy of segregated apartheid hospital planning. As a result, the designated maternal–newborn specialist facility was *"at capacity probably 364 days of the year"* (S, D1), and unable to accept referrals. In contrast, the district hospital in another sub-district of D1 established its own specialist obstetric and paediatric services, including a neonatal intensive care unit. While compensating for weaknesses at a higher level, this district hospital lacked key support functions, such as a blood bank, normally associated with specialised services. These imbalances necessitated considerable day-to-day problem solving and brokerage by clinicians and programme managers across the platform. As recounted by a senior manager in this hospital: *"We've even swopped babies, you know, ... that's an ongoing thing and Dr X [DCST paediatrician] really assisted us in this regard being a liaison between us and [referral hospital] ..."* (S, D1).

In the two rural districts (D2 and D3), there was a perceived mismatch between the distribution of skilled midwives and uncomplicated maternal deliveries, specifically between (under-utilised) 24 h CHCs and (overworked) district hospitals. The chief executive officer (CEO) of a district hospital described the efforts and persuasion required to shift this pattern: *"[CHC name] has got two delivery rooms, their challenge was water...I assisted them and said 'now deliver'. And they said 'hey, we have not delivered for so long we can't remember.' I said 'come to the hospital and refresh.' But then they said 'look, when do I come to the hospital because in the clinic we are short staffed?' There are all those dynamics..."* (S, D3). A more rational distribution of staff and services was made difficult by fragmented human resource pools and reporting lines between PHC and hospital services. High turnover of junior community service doctors at these hospitals also posed a challenge (discussed further under outreach).

Respondents in these districts highlighted how relatively small but critical bottlenecks in supplies had ripple effects on the referral pathway. For example, district hospitals had established neonatal high care units, including staffing, equipment (e.g., incubators and continuous positive airway pressures (CPAP) machines), essential drugs (e.g., surfactant for pre-term infants), and dedicated champions. However, one unit was without medical air and another without replacement piping for the CPAP machine for some months, creating referral overloads of pre-term infants to their respective regional hospitals. This was also a source of demotivation: *"I feel like the disappointments in the system, the struggles logistically*

*. . . has tapped the energy of many people and so essentially they are in a space where they just don't really care that much anymore"* (D, D3).

### 3.1.3. Functioning Emergency Medical Services

Overall perceptions of the referral system correlated with EMS functioning in each district. In urban D1, although not without challenges (highlighted in Supplementary Table S1), inter-facility transfers by EMS were considered largely unproblematic by both PHC and hospital clinicians: *". . .we can transfer a patient from [CHC name] MOU [midwife obstetric unit] to [district hospital name] and the patient is there within an hour after discussion, and the same goes for transfers to [tertiary hospital name]"* (S, D1). This is borne out in the data on response rates and availability of vehicles.

In contrast, in D3, EMS was "massive trouble" (S, D3) at all levels of the system: "[the regional hospital] [will] accept the patient . . . but the patient might definitely leave after six or eight hours and that is so frustrating, it creates a lot of headaches" (S, D3). In the PHC system, the wait for an ambulance could "go up to three hours after you have called. . . sometimes it does not come at all. . ." (S, D3). Acute shortages of ambulances (evident in very low ratios per population) and skilled personnel were compounded by poor triaging and prioritising at the central provincial call centre, unable to distinguish between a genuine emergency and people calling the ambulance "as a taxi . . . to town" (S, D3). Ambulances were not stationed at or near health facilities where district and sub-district players could develop informal relationships with EMS managers.

Despite a shortage of ambulances and skilled staff, the EMS in D2 was considered reasonably functional against a backdrop of active efforts by provincial authorities to resolve challenges. Ambulances were stationed at facilities (even if the call centre remained central), and EMS managers became assimilated into the informal systems of communication and accountability in the district: *". . .we've got a WhatsApp group here. . . If I have called the ambulance and there is a delay, she posts the message to the WhatsApp, . . . and whoever sees the message, it's connected to the district office. . . even the district manager, you will see they're responding 'I have called the EMS manager to assist'. Yeah, we will communicate in that way, [so] that there is no maternal issue that is not attended. And immediately the EMS manager saw that there is something downwards, his people are not attending to it, he will push because he knows when they go for a meeting, it will be tough"* (D, D2).

Interviewees were in favour of ambulances stationed at district and regional hospitals for inter-facility transfers of emergencies. One district hospital manager further indicated that (contrary to national regulations) these ambulances would not require skilled EMS personnel, often a referral stumbling block, since a hospital staff member could accompany the patient and return to the hospital in the same ambulance.

### 3.1.4. Communication and Collaboration

Communication and collaborative relationships were considered the essential 'software' [27] of smooth referral systems. As indicated by a clinical manager: *". . .we communicate frequently and we make sure that we keep in touch . . . because we know that we are the ones who need the services"* (S, D3). Dense networks of communication, straddling a variety of interfaces, were evident in all three districts, which were made possible by common platforms (in particular WhatsApp) and new modes of convening since the COVID-19 pandemic. In one district, the Director of PHC Services was in three WhatsApp groups involving a progressively wider circle of players above and below her in the district. Interactions in these groups were key to everyday problem solving, as described for EMS above, and produced a general shift from the formal to the informal and from face-to-face to virtual modes of communication and decision-making. As recounted by one manager: *"Initially we never had the informal meetings. We only had formal meetings. . .. I think people are getting more used to those things. So it is easier to manage than when we're waiting for formal meetings, because formal meetings might be monthly, might be biweekly, it takes longer. But informal is better because you do it as and when it is needed . . . it addresses the problem immediately"* [D, D2].

While there was a move towards technology-enhanced environments, and cellular communication and Wi-Fi in particular, access to these technologies was not universal. A key challenge, lying beyond the procurement capabilities of the health system, was the unavailability of mobile networks in deep rural areas, posing significant challenges for emergency referrals in the absence of alternatives (such as landlines and radios). Moreover, the technologies in use, while enabling new communication networks, remained at a basic level. In one district, a nationally developed referral and communication application (called 'Vula') was being trialled, but beyond this, more advanced forms of telehealth were not evident.

Communication networks were supported by semi-formal local structures established to enable collaboration across referral interfaces. In both D1 and D3, clinical managers were part of regular medical forums, playing a variety of instrumental (e.g., sharing of specialised resources) and mentoring roles. In D1, this forum met monthly, discussing *"whatever problems we have in the institutions and there's a lot of fruitful things that came [of that]... in our various discussions we've noted that psychiatric services is a problem. And now we're trying to establish an outreach from [psychiatric hospital] for their specialist to come to our hospital to assist with the care of our mental health care patients... it's also a good platform to integrate services"* (S, D3). Similar problem solving fora between nursing service managers and between PHC facilities and the local hospital were also reported.

However, these various coordination mechanisms were patchy and relied on champions, tended to follow professional lines, and did not necessarily involve all the relevant players required for authoritative decision-making; they were also not always able to overcome structural silos, in particular, the separation of PHC, district hospital and EMS services at sub-district levels. Collaborative relationships at these interfaces remained uneven, and as indicated earlier, created inefficiencies in services, staffing, and accountabilities in the referral chain. For example, *"if my perception ... working in the higher level is that the lower level staff... did not actually manage the patient well or they just attempted to move it on to the next level of care, how do you then address that because the person who is in charge of the hospital doesn't have any jurisdiction over the clinic staff?"* (D, D3). Similarly, a vertical EMS reporting line to the province undermined relationships and coordination at district level: *"EMS is a little island ... EMS comes when they have got challenges and then they expect intervention from the district, but when it also suits them [they say] 'we don't report to the district so the district can't tell us what to do'"* (D, D1). Senior managers in two provinces, P1 and P2, recognised these as core governance and system design challenges and introduced, or were in the process of introducing, appropriate system reforms (discussed further under governance).

*3.2. Outreach Systems*

3.2.1. Models of Outreach

MNH outreach in the three districts consisted of a combination of long-established programme managers (nursing cadres) at district and sub-district levels, outreach from facility clinicians (medical and other) in a cascade model spanning tertiary to PHC services, and the DCSTs based in the district office (Supplementary Table S2).

Although popular amongst frontline providers, the DCSTs were being phased out in the three provinces. At the time of data gathering, only D3 still had a sizeable DCST, consisting of five members. D1 had a district paediatrician and shared an obstetrician with a neighbouring district, while in one province (P2), the DCSTs were disbanded and replaced with a system of specialist outreach from regional hospitals. This was conducted, according to a senior provincial manager, because *"when patients are supposed to be transferred to a hospital which need the resources in terms of specialists, the specialists are not there ... we have a gynaecologist in the district office ...but at the regional hospital where the ICU bed is ... we don't have a specialist ... that's why they took a decision to say specialists in the regional hospital ... will do outreach from the regional hospital to the district hospital, not from the district office"* (Pr, D2). Medical specialists were appointed at the five regional hospitals *"...to oversee the entire clinical operations ... in their catchment area. And what we wanted to see happen here is that*

*the clinical leadership actually takes ownership and accountability for all clinical processes—case management, case referral, down referral, out referral, clinical support for them, clinical support meaning outreach support for clinical care and governance, reviewing data as a unit, responding to data as a unit*" (Pr, D2). Two provincial coordination, support and oversight mechanisms, referred to as obstetrics and neonatal response teams, were established by tertiary level clinicians, bringing together regional clinicians, programme managers, and key support functions (blood bank, EMS, pharmacy).

Provincial managers in another one of the provinces (P1) also viewed this as the most rational approach: "if we want a system that works well, our regional and tertiary facilities should have clinicians that have structured outreach programmes. Now the DCST is . . . a compromise because we didn't have enough people to do that" (Pr, D1). In this province, the remaining DCSTs were increasingly being drawn into provincial strategic and system (re)design roles.

An additional challenge was that DCSTs were not planned with the programme managers in mind and there was duplication of roles, particularly in the nurse-based PHC system. As one programme manager recounted: *"to me it is almost same, DCSTs they mentor and train, and make sure if policies are implemented, but most of the time it's done by a programme manager"* (D, D1). An implicit hierarchy between the DCSTs and MNH programme managers furthermore undermined the *"ownership and agency"* (Pa, D2), of the latter.

Although P2's cascade model and central response teams were considered the most efficient and sustainable design for outreach, the approach was still in the early stages of implementation and faced a number of challenges. This model required, firstly, significant mindset shifts amongst specialists: *". . .when the DCSTs were disbanded, obviously we went to the specialist guys, this is the new model, and they looked at us as though we were crazy that we were saying that they must go and do what? What about the patients in their hospital? So, they just didn't get the concept that problems at the PHC become your district hospital problems. Problems at the district hospital become your regional hospital problems. Regional hospitals automatically become your tertiary hospital problems"* (Pa, D2); secondly, availability of resources for travelling in the district, which were yet to be provided; and thirdly, a system of oversight with clear expectations and answerability: *"If nobody's going to ask you, did you do outreach, if nobody's even worried about [it] . . . how are you going to be reimbursed, you're just going to sit back and not do it"* (Pa, D2).

In the light of this, frontline clinicians, especially in district hospitals expressed a preference for the dedicated and *"hands on"* (S, D2) clinical governance and support provided by the DCSTs, and their disbandment in D2 was experienced as a loss. Paradoxically, the MNH outreach system in D3, which operated without a provincially formulated or supported outreach strategy, but which had a combination of a strong DCST, programme manager, and regional hospital, was perceived the most favourably of the three districts.

### 3.2.2. Outreach Roles

Table S2 summarises the outreach roles of clinical care, training/mentorship, and audit/M&E in the three districts. Regional hospital specialists and DCSTs generally provided clinical outreach in district hospitals, but bar a few exceptions, there was no clinical outreach from district hospitals to PHC for maternal–newborn services. Complex cases were referred up to 'high risk' clinics. This was considered a key gap—for example, in support for the identification and management of hypertension antenatally—and was in contrast to the more developed outreach to PHC clinics for other disease programmes (notably HIV and TB).

All three districts had active MNH in-service education programmes based on nationally recognised courses. Programme managers (with support from DCSTs) steered the planning and organisation of training through a variety of structures and processes: regional training centres (D1), partner initiatives (D2, D3), and the obstetric and neonatal response teams (D2). In-service training appeared more anchored in the planning and

management routines of D1 and D2 than in D3, where continuing education was *"not departmentalised"* (D, D3), and relied on ad hoc arrangements, described by one manager as *"normalised anomaly"* (S, D3). Provincial clinical specialist coordinators in this province who were supposed to oversee the outreach system were no longer in place. There were gaps, for example, in the induction of new community service doctors in surgical and anaesthetic skills to conduct caesarean sections, referred to by one interviewee *as "a public health disaster waiting to happen"* (D, D3). Clinical skills development in this district was de facto delegated to clinical managers in district hospitals *"placing a lot of reliance on one individual to be multi-skilled"* (D, D3).

A mix of national, provincial, and district MNH audit tools and processes were in use in the districts, and were understood to be a core, clinical governance function of programme managers and specialist teams. Notwithstanding the interruptions of COVID-19 lockdowns, many facilities had perinatal mortality review meetings, feeding data into an information system referred to as the Perinatal Problem Identification Programme (PPIP). However, PPIP meetings were attended by clinicians who did not have decision-making authority or control over resources: *"...they [DCST] will go and do some audits in facilities and they will come up with recommendations and they will send to CEOs and to the executive in our office. But you might not see those things implemented. Because maybe there is a question to say 'who are they?'. And how far their role? Can they instruct the CEOs, the medical managers to make sure that things are implemented?"* (Pr, D3)

In one province (P2), where reducing maternal mortality was a stated political priority, senior provincial managers mandated the establishment of monthly performance monitoring and response forums (PMRFs) in each sub-district, bringing together providers (medical, nursing, and allied), as well as line and support staff in the immediate catchment area to address bottlenecks. The sub-district PMRFs would then meet together quarterly at the district level. Similar processes were underway in a second province (P1). In one sub-district of D1, the Mphatlalatsane Project introduced a structure referred to as the Monitoring and Response Unit (MRU), where clinicians and managers from the district hospital and feeder facilities met on a regular basis to review maternal–neonatal outcomes. As expressed by a hospital clinical manager, achieving health outcomes: *"does require the whole team, the whole district level platform if I can put it that way to function as a team and I think that the promotion of such a teamwork and effort will definitely be to improve a lot of things. Be that the referral system, quality of care, be that clinical skills... there's a lot that we can learn from primary health care, there's a lot that primary health can learn from us"* (S, D1). The plan was to extend this approach to other sub-districts.

Districts also conducted audits of, and reported, maternal deaths to a National Committee on Confidential Enquiries into Maternal Deaths; and MNH was included in general monthly and quarterly reviews at sub-district and district levels, respectively. In addition, there were provincially designed tools and processes, such as the maternal health care standards and neonatal quality 'Facility Assessment Tools' in P2. While referral processes were often identified as a key factor in maternal and perinatal deaths, there was little formal monitoring of these systems. EMS turnaround times were included in the National Indicator Data Set, but were inconsistently reported.

### 3.3. The Governance and Leadership Context

In P2, the PMRFs were introduced in the context of a wider governance redesign referred to as the geographical service delivery (GSD) model. In this model, *"if you are a CEO of a hospital . . . that whole catchment area should be the responsibility of the CEO, . . . similarly the clinical manager is responsible for clinical services in that area and all the outcomes. . . So even the issue of . . . HR responsibilities, you would want them to be accountable and responsible . . . for who will be based at the hospital, even the rotation of staff could be around that similar geographical area. Because now what happens is when you are recruiting, you would recruit for the clinic, but the hospital is also saying they have a shortage . . . if you know that, okay, actually I'm going to work in this geographical service area, so even if I am at a clinic, I could be called to the maternity ward in*

*the hospital...*" (Pr, D2). The cascade approach to outreach and the provincial obstetric and neonatal response teams were also aligned with the GSD vision and service reorganisation. At the time of the research, the GSD model was still being introduced, but was already perceived to be making a difference: "*since we came in with the geographical arrangement, our relationships have improved a lot*" (S, D2).

Similarly, in P1, the establishment of MRUs was part of a growing recognition of the need to strengthen the sub-district as the core service delivery and governance unit: "*at the end of the day your sub districts should actually be the heart of everything... That is actually where you should have your resources to make it happen*" (D, D1). The province was also embarking on "*service delivery optimisation*", which in D1 included repurposing a specialised tuberculosis care hospital into a general district hospital with maternal–child services.

Such new thinking and service delivery re-arrangements were not evident in the third province, P3. While the "*noises that were made*" (S, D3), including by the Public Protector (a national oversight body), on neonatal services at the regional hospital in D3 reportedly resulted in a facility upgrade, the provincial sphere was experienced as uninvolved and/or unable to resolve major structural challenges. There was a high turnover in senior provincial leadership, with key positions (including the Head of Department) vacant at the time of the research, and a perceived loss of strategic and technical skills (described as "*the centre not holding*" (S, D3)), alongside an intense politicisation of the public service, where managers were easily "*sacrificed*", and were "*more afraid of unions than their superiors*" (Pa, D3).

All three districts had a high turnover of leadership teams in health facilities, especially hospitals, which was reflective of fractious social and labour relations in these institutions. Positions became filled by "*...young, young, young clinical managers and young, young, young CEO's that don't know what to do*" (Pa, D2) who received little induction and mentorship.

## 4. Discussion

Table 3 summarises the enablers and constraints of MNH referral and outreach systems in the three districts. The similarities and contrasts (from reasonable to poor) in participants' experiences of referral and outreach shed light on the factors shaping these functions and more broadly on the management of interdependencies in local health systems.

One of the most striking features of the three districts was the uniqueness of their provincial contexts, despite all three being affected by the wider crisis of (mal)governance and of the politicisation of provincial health systems [28]. An emergent [2] orientation towards recognising interdependence was evident in the strategies and system redesign of P2 (with the adoption of the GSD, regional specialists and PMRFs), and to some extent in the P1, with moves towards an integrated sub-district model. Over the last two decades, community-based services, PHC, district hospitals, health programmes, and EMS evolved as vertical siloes in separate reporting lines to higher levels. However, improved MNH rests fundamentally on the coordinated actions of these players at a sub-district level [16]. Proactive, adaptive organisational strategies [2] to overcome system fragmentation, evident in two of the provinces, are mirrored in global thinking on the importance of 'Networks of Care', which "purposefully and effectively interconnect service delivery touch points within a catchment area to fill critical service gaps and create continuity in patient care." [13].

While the enabling frameworks of provincial authorities can positively impact lower levels, they remain in tension with wider governmental frameworks of reporting and accountability focused on individual units where measurement and attribution are possible (in contrast to diffuse collective responsibility). Regional (in D1 and D3) and tertiary hospitals report in separate 'budget programmes' to provincial structures, and unless specifically mandated, have little incentive to cooperate with each other or district players. The National Department of Health is itself organised into vertical programmes and functions, and has a natural inclination towards siloed engagement and technical interventions, introduced through ring-fenced budget allocations (such as the DCSTs). This speaks to a mismatch of perspectives and imperatives. In a federal system such as South Africa, the disjuncture in decision-making between national and provincial spheres hampers the

"continuous feedback between small-scale and large-scale perspectives" regarded as "the essence to achieving an efficient and effective health system" [3].

**Table 3.** Enablers and constraints of referral and outreach systems in the three provinces/districts.

| Dimensions | Province 1 (P1), District 1 (D1) | D2 Province 2 (P2), District 2 (D2) | Province 3 (P3), District 3 (D3) |
|---|---|---|---|
| Enablers | • Active processes of system redesign, 'service delivery optimisation', and strengthening of the sub-district<br>• Provincial procurement of ambulances<br>• Presence of system mediators/brokers<br>• Clinical manager network across hospitals, and between PHC and DH in one sub-district<br>• Relational ecosystems and organisational cultures support functioning of referral systems<br>• Nodes of skilled and committed leadership | • Introduction of new 'geographical service delivery model', entailing<br>  ○ Service delivery coordination between PHC and DH<br>  ○ New accountability relationships and coordination between line, clinical, and support managers<br>• Provincial political commitment and active stewardship, incl. addressing bottlenecks; launch of maternal standards; appointment of specialists and mandating cascade model of outreach<br>• District leadership<br>• Presence of system mediators/brokers<br>• New forms of communication (WhattsApp and virtual meetings) | • Stable and supportive district management team<br>• Nodes of system strength, including regional hospital providing an enabling role for the district<br>• Presence of skilled clinical specialists and programme managers playing active brokerage roles and supporting innovation<br>• Dense networks of informal communication and problem solving<br>• Vula App being implemented |
| Constraints | • Lack of formal sub-district structures in metro, fragmented reporting lines<br>• Problematic system design inherited from the past<br>• Resource constraints<br>  ○ Acute budget shortages<br>  ○ Lost key middle managers over the COVID period<br>  ○ COVID contracts terminated<br>• Narrow district decision space<br>• Politicisation of services, work stoppages<br>• No coordination platforms for MNH | • Not enough beds and capacity in referral hospitals<br>• Inefficient distribution of skilled staff and services at sub-district level creating 'phantom shortages'<br>• Unstandardised and variable supply chains<br>• High turnover of clinical managers and facility leadership in district and regional hospitals<br>• Loss of provincial programme capacity<br>• Poor mobile network availability | • Fragmented reporting lines between PHC and hospitals at sub-district level<br>• High turnover of senior managers and little active stewardship of district services at provincial level, incl.<br>  ○ Service delivery and referral system (re)-design<br>  ○ Coordination between district and regional/tertiary hospitals<br>  ○ Addressing critical weaknesses in EMS and unresolved infrastructural and supply chain challenges<br>• Clinicians 'not taken seriously', power to allocate resources<br>• Politicisation of services, work stoppages<br>• Weak provincial capacity 'centre not holding' |

National and provincial policy and frameworks on referral offer important guidance, but as was evident, they require further deliberation, adaptation, and agreement between players in specific referral pathways, with their own facility configurations, capacities, and catchment areas. Communication and collaboration in referral pathways are thus key, and

in at least one study district was greatly enabled by the near universal and rapid adoption of informal, real-time mechanisms such as WhatsApp, and the general leap-frogging in the use of technology during the COVID-19 pandemic.

MNH outreach systems build relationships and create enabling environments for smooth referrals. They function best when ongoing skills development plans and processes are an integral part of core district and facility systems; and audit and review processes combine clinical and managerial decision-makers to address resource challenges. The change in MNH outreach from dedicated specialist teams towards more integrated, cascade models required new resource allocations and accountabilities, but also significant mindset shifts. Health professionals are trained in silos, focused on individual clinical acts, and tend towards what Frenk et al. [29] referred to as 'tribalism', that is, "the tendency of the various professions to act in isolation from or even in competition with each other" [29]. In the study districts, spontaneous forms of self-organisation often evolved along professional lines, clinical governance processes were unconnected to line management, and new forms of specialist outreach (DCSTs) tended to be layered on and parallel to the nurse-based programmatic structures. However, there was also evidence of system innovators, whether external partners or internal system brokers, who were able to challenge atomised functioning, and of providers and managers embracing systems thinking [30]. They were catalysts of 'integrative' functioning across silos, professions, and hierarchies [31]. Such cadres could play an important role in future district health system development as 'hybrid' managers, "able to embody, translate and mediate" clinical and managerial logics [32].

Finally, two leverage points, "places within a complex system, where a small shift in one thing can produce big changes in everything" [33], for improving the MNH referral system were identified. The first of these is strengthening EMS, often a critical stumbling block in the MNH referral system [11,13], which faces many challenges, including shortage of skilled personnel, ambulances and other resources, delays in inter-facility transport, inadequate emergency care, and insufficient monitoring and management of patients during transport [10,11]. An experiment with dedicated maternity ambulances for inter-facility transfers in one province led to a 45% reduction in maternal deaths [34]. However, this innovation was not sustained in this province or scaled up elsewhere, and an evaluation conducted in three provinces in 2018 found that only one-quarter of EMS stations had dedicated maternal–newborn ambulances [35]. Although such vertical interventions may not be sustainable, strategies to better manage the dependence on the EMS system, for example, co-location [3] of services at health facilities, were widely supported. The second leverage point is in strengthening the procurement of essential supplies and equipment in maternal and newborn units. Relatively small bottlenecks within individual health facilities often had major knock-on effects on referral pathways and could be resolved at district and even sub-district levels.

*Study Limitations*

The focus on maternal–newborn referral inevitably puts a spotlight on the management of emergencies, which may obscure the functioning of the more common and less dramatic movements of people through care pathways. Seen through the lens of chronic illness (whether HIV or diabetes) or from a life course perspective, the analysis would reveal different players and system interdependencies and would also bring the relationship between users, citizens, and the PHC system more clearly into focus.

## 5. Conclusions

This paper fills a gap in understanding of district maternal–newborn referral and outreach systems, and through this lens, the management of local health system interdependencies. The analysis surfaced key elements of district health system functioning, specifically in the connectedness of system elements, and the relational ecosystems, system brokers, systems thinkers, and governance frameworks that enable connectedness. A systems analysis focused on interdependent processes provides the conceptual tools for better

understanding the DHS performance, while identifying leverage points for catalysing wider change in health systems characterised by complexity.

**Supplementary Materials:** The following are available online at https://www.mdpi.com/article/10.3390/systems11090462/s1, Table S1: MNH referral systems in the three study provinces/districts; Table S2: MNH outreach systems in the three provinces/districts.

**Author Contributions:** Conceptualization, H.S. and S.M.; formal analysis, H.S., S.M., W.O. and T.C.; funding acquisition, T.C.; investigation, H.S., S.M. and W.O.; methodology, H.S. and S.M.; validation, H.S., S.M., W.O. and T.C.; writing—original draft, H.S. and S.M.; writing—review and editing, H.S., S.M., W.O. and T.C. All authors have read and agreed to the published version of the manuscript.

**Funding:** This work was supported by the South African Department of Science and Innovation/National Research Foundation South African Research Chair's Initiative (grant number 98918); and by the ELMA Philanthropies via the South African Medical Research Council to UWC (grant number 46241).

**Data Availability Statement:** Tables S1 and S2 provide detailed inventories of qualitative themes.

**Acknowledgments:** The authors are deeply grateful to the Mphatlalatsane project designers, implementing partners, and provincial, district and sub-district interviewees for so readily sharing their insights, experiences and documentation.

**Conflicts of Interest:** The authors declare no conflict of interest. The funders had no role in the design of the study; in the collection, analyses, or interpretation of data; in the writing of the manuscript; or in the decision to publish the results.

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
