# Peer review of "Managing Local Health System Interdependencies: Referral and Outreach Systems for Maternal and Newborn Health in Three South African Districts"

_systems, doi:10.3390/systems11090462_

Round 1

Reviewer 1 Report

GENERAL COMMENTS

Excellent study, from design to execution.

Congratulations to the authors for generating such high quality and very helpful knowledge and insight, with factual information as well as live experiences that are particularly compelling for both health service delivery providers and policy makers/managers.

Just one point, for consideration: the concept of “outreach” is adequately defined and referenced as including “a variety of activities that combine aspects of clinical care, continuing professional education and clinical governance” (Introduction, para 4). Yet, this definition is somewhat novel to me (compared to a definition of outreach that I am more familiar with as a service delivery strategy that actively takes services out from health facilities and brings them closer to communities/population groups, including through mobile strategies). Hence, it may be helpful to provide one or two examples of “outreach” according to the authors’ approach and vision.

SPECIFIC COMMENTS

As lines in the manuscript are not numbered, I refer to paragraphs/sub-paragraphs.

Introduction

Para 1: “(p196)” is not needed, may be deleted.

Figure 1: “p7” is not needed, may be deleted.

Study setting

Para 3: “(p23)” is not needed, may be deleted.

Discussion

Para 3: “(p21)” is not needed, may be deleted.

Para 4: “(p1007)” is not needed, may be deleted.

Para 6: “(p1)” and “(p74)” are not needed, may be deleted.

Para 7: “(p1)” is not needed, may be deleted.

Author Response

Many thanks for your positive feedback!

Comment:

Just one point, for consideration: the concept of “outreach” is adequately defined and referenced as including “a variety of activities that combine aspects of clinical care, continuing professional education and clinical governance” (Introduction, para 4). Yet, this definition is somewhat novel to me (compared to a definition of outreach that I am more familiar with as a service delivery strategy that actively takes services out from health facilities and brings them closer to communities/population groups, including through mobile strategies). Hence, it may be helpful to provide one or two examples of “outreach” according to the authors’ approach and vision.

Response: Page 3, line 88 specifies that outreach refers to clinical rather than community outreach

SPECIFIC COMMENTS

As lines in the manuscript are not numbered, I refer to paragraphs/sub-paragraphs.

Introduction

Para 1: “(p196)” is not needed, may be deleted.

Figure 1: “p7” is not needed, may be deleted.

Study setting

Para 3: “(p23)” is not needed, may be deleted.

Discussion

Para 3: “(p21)” is not needed, may be deleted.

Para 4: “(p1007)” is not needed, may be deleted.

Para 6: “(p1)” and “(p74)” are not needed, may be deleted.

Para 7: “(p1)” is not needed, may be deleted.

Response: Page numbers have been deleted

Reviewer 2 Report

Dear author(s),

The manuscript is very interesting and accurately written but I would kindly ask the author(s) to insert some information about the study population in terms of years of employment in the institutions and if the had middle or top management positions. It is very interesting to have different perspectives.

I did not find any  Conclusion section. Please include one. 

Thank you!

Good luck!

Author Response

Comment:

The manuscript is very interesting and accurately written but I would kindly ask the author(s) to insert some information about the study population in terms of years of employment in the institutions and if the had middle or top management positions. It is very interesting to have different perspectives.

Response:

Unfortunately we did not record data on duration in employment. We have clarified their hierarchical positions on page 6 (lines 173-77).

Comment:

I did not find any  Conclusion section. Please include one. 

Response:

We have added a conclusion (page 17, lines 644-672)

Reviewer 3 Report

This paper reports on a qualitative study of referral and out-reach systems for MNH care in three districts of three South African provinces, and what these systems reveal about the local management of health system interdependencies.

The key issue identified – para 1 p 13 – is that ‘improved MNH rests fundamentally on coordinating actions of players at subdistrict level’

Overall the paper has a clear structure, and the methods, analysis and interpretation of the results are clearly and adequately described.

My main comment relates to the focus on the provision of EMS services, which could be considered the ‘supply side’, and the neglect of information on the ‘demand side’, such as the population characteristics (birth rate, SES), MNH service function indicators (ANC1, ANC 4, SBA), MNH workforce availability and MNH outcomes – IMR, NMR, MMR. This information would provide some indication of the demand for referral or Emergency services. Information on the available workforce such as midwives per population, delivery beds per population, would also provide information on the capacity of the obstetric services to address this demand. Both these factors then contribute to the demand for EMS services.

I appreciate that the authors face limitations of space, and the paper is already quite dense in information. The supplementary tables provide some information – such as number of ambulances per population, but little information on the characteristics above. Perhaps this information could be included in the supplementary tables, with a reference in the text.

Author Response

Comment:

My main comment relates to the focus on the provision of EMS services, which could be considered the ‘supply side’, and the neglect of information on the ‘demand side’, such as the population characteristics (birth rate, SES), MNH service function indicators (ANC1, ANC 4, SBA), MNH workforce availability and MNH outcomes – IMR, NMR, MMR. This information would provide some indication of the demand for referral or Emergency services. Information on the available workforce such as midwives per population, delivery beds per population, would also provide information on the capacity of the obstetric services to address this demand. Both these factors then contribute to the demand for EMS services.

I appreciate that the authors face limitations of space, and the paper is already quite dense in information. The supplementary tables provide some information – such as number of ambulances per population, but little information on the characteristics above. Perhaps this information could be included in the supplementary tables, with a reference in the text.

Response:

We agree that our analysis does not capture the ‘demand’ for EMS services. However, inter-facility MNH transfers are just one element of a much wider scope of services provided by EMS (much of which is public facing), and differences in demand between districts is unlikely to have been the cause of the variable experiences

Beyond this, we agree the analysis would benefit from a quantitative assessment of the match between supply and demand. The paucity of data on referral systems themselves is also a major limitation. As part of the study setting (Table 1, page 5), we have reported the ‘UHC index’ for each district. This index  was recently developed in South Africa and is a composite of supply and demand indicators, including reproductive, maternal and child health with health service (staffing and beds) indicators (https://www.hst.org.za/publications/District%20Health%20Barometers/District+Health+Barometer+2018-19+Web-255-276.pdf). All three districts were below the national average for this index, but clustered close to each other.

Data on midwife availability would have provided useful insights on facility capacity but are not routinely reported; and while adding data on mortality and ANC coverage may be valuable contextual information, it is unclear how they could add to the analysis of referral and outreach (as cause or effect?) .